# Onabotulinumtoxina in the Prevention of Migraine in Pediatric Population: A Systematic Review

**DOI:** 10.3390/toxins16070295

**Published:** 2024-06-28

**Authors:** Artemis Mavridi, Aine Redmond, Paraschos Archontakis-Barakakis, Petya Bogdanova-Mihaylova, Christina I. Deligianni, Dimos D. Mitsikostas, Theodoros Mavridis

**Affiliations:** 1First Department of Pediatrics, School of Medicine, “Aghia Sofia” Children’s Hospital, National and Kapodistrian University of Athens, 11527 Athens, Greece; a_mavridi@hotmail.com; 2Department of Neurology, Tallaght University Hospital, D24 NR0A Dublin, Ireland; airedmon@tcd.ie (A.R.); petya.mihaylova@tuh.ie (P.B.-M.); 3Redington-Fairview General Hospital, Skowhegan, ME 04976, USA; p.archontakis.barakakis@gmail.com; 41st Department of Neurology, Eginition Hospital, Medical School, National and Kapodistrian University of Athens, 11528 Athens, Greece; cdchristina@gmail.com; 5Neurology Department, Athens Naval Hospital, Deinokratous 70, 11521 Athens, Greece

**Keywords:** OnabotulinumtoxinA, pediatric migraine, PREEMPT, migraine treatment, chronic migraine, preventive treatment

## Abstract

Migraine is a leading cause of disability worldwide, yet it remains underrecognized and undertreated, especially in the pediatric and adolescent population. Chronic migraine occurs approximately in 1% of children and adolescents requiring preventive treatment. Topiramate is the only FDA-approved preventative treatment for children older than 12 years of age, but there is conflicting evidence regarding its efficacy. OnabotulinumtoxinA is a known and approved treatment for the management of chronic migraine in people older than 18 years. Several studies examine its role in the pediatric population with positive results; however, the clear-cut benefit is still unclear. OnabotulinumtoxinA seems not only to improve disability scores (PedMIDAS) but also to improve the quality, characteristics, and frequency of migraines in the said population. This systematic review aims to summarize the evidence on the efficacy, dosing, administration, long-term outcomes, and safety of onabotulinumtoxinA in pediatric and adolescent migraine. Eighteen studies met the eligibility criteria and were included in this review. The mean monthly migraine days (MMDs), decreased from of 21.2 days per month to 10.7 after treatment. The reported treatment-related adverse effects were mild and primarily injection site related and ranged from 0% to 47.0%. Thus, this review provides compelling evidence suggesting that OnabotulinumtoxinA may represent a safe and effective preventive treatment option for pediatric migraine.

## 1. Introduction

Migraine is a primary headache disorder affecting approximately 11% of the pediatric population [1]. According to the International Classification of Headache Disorders, chronic migraine is defined as headache attacks that occur more than 15 days per month for more than 3 months, which, on at least 8 days/month, has the features of migraine headaches [2]. Like in the adult population, chronic migraine is reported to affect 1% of children and adolescents [3]. The quality of life is severely affected in this population, resulting in school absence, reduced productivity (presenteeism), and restriction of social activities [4]. Furthermore, children with migraine present more somatic complaints and are more prone to anxiety disorders, depression, and attention deficit problems [5]. The management of migraine includes acute and preventive treatments in addition to behavioral and lifestyle changes. Initial acute treatment options, recommended by many scientific societies, include oral analgesics such as acetaminophen, ibuprofen, and naproxen [6,7]. Triptans are also used in pediatric patients for the acute management of migraines, and four of them (sumatriptan, almotriptan, zolmitriptan, and rizatriptan) are approved by the Food and Drug Administration [8] and the European Medicines Agency [6,7]. Children often respond sufficiently to acute treatment measures and do not require preventative therapy [9].

The goal of preventative therapy is to reduce the frequency and intensity of migraine attacks and to improve the level of the associated disability which is assessed by the Pediatric Migraine Disability Score (PedMIDAS), a six-question tool that evaluates the influence of migraine episodes on both school attendance and functionality, as well as the participation in extracurricular activities [9,10]. While topiramate is the only drug approved by the U.S. Food and Drug Administration (FDA) as a preventative treatment for chronic migraine in adolescents older than 12 years, most randomized controlled trials fail to demonstrate superiority over placebos [8,9,11]. Furthermore, recent directives from the European Medicines Agency (EMA) have contraindicated its use in women of childbearing potential unless they are utilizing highly effective contraceptive methods [12,13,14,15,16,17]. Additional preventative pharmaceutical options that are commonly used in pediatric chronic migraine include tricyclic antidepressants [11], non-selective beta-adrenergic receptor antagonists [18], and other antiepileptic medications (sodium valproate) [9,19,20]. However, the existing randomized, placebo-controlled studies show that there is no statistically significant reduction in the frequency or severity of migraine episodes using the aforementioned treatments [11,18,21]. Flunarizine, a calcium channel blocker, appears to have a positive effect on migraine attack frequency in older studies, but novel studies are needed to confirm this finding [22]. Thus, new targets for anti-migraine treatments are needed for this population.

The first open-label study investigating OnabotulinumtoxinA’s role in adults with migraine was conducted in the early 2000s [23]. Several years later, the Phase III Research Evaluating Migraine Prophylaxis (PREEMPT) trials established the efficacy and safety of OnabotulinumtoxinA in chronic migraine in adults [24,25,26]. OnabotulinumtoxinA is the only botulinum toxin with FDA approval for the treatment of chronic migraine in adults, a designation that it received in 2010 [27]. Other botulinum toxins, including abobotulinumtoxinA, incobotulinumtoxinA, and rimabotulinumtoxinB, lack FDA approval for this specific indication. In pediatric patients, the evidence is scarce. The purpose of this review is to highlight the efficacy along with the safety of OnabotulinumtoxinA in children and adolescents with chronic migraine.

## 2. Results

Our systematic search initially identified 676 records. After removing the duplicates, 541 articles were screened based on the title and abstract, leading to a full-text review of 169 articles. Ultimately, 14 studies met our eligibility criteria and were included in this review. The PRISMA flow diagram illustrates the study selection process (Figure 1).

### 2.1. Study Characteristics

The studies included in the analysis consisted of two randomized controlled trials (RCTs) [28,29], nine retrospective analyses [30,31,32,33,34,35,36,37,38], and three case series [39,40,41] involving a total of 476 pediatric patients. All the studies included patients with chronic migraine. To define chronic migraine, most studies adopted the latest ICHD-3 criteria [30,33,34,35,37], whereas three studies employed ICHD-2, including both RCTs [28,29,32]. In the remaining studies, the criteria utilized to define chronic migraine were unspecified [31,36,38,39,40,41]. Most of the studies followed the PREEMPT injection and dosing protocol, which consists of intramuscular (IM) administration of OnabotulinumtoxinA 155 IU in 31 fixed sites in seven distinguished muscle areas (frontalis, temporalis, occipitalis, cervical paraspinal, trapezius, corrugator, and procerus muscle) every three months. An additional 45 IU could be administrated using a “patient-specific follow-the-pain” strategy [30,31,33,34,37,38]. Both RCTs had a stricter dosing protocol. More specifically, Shah et al. offered a 155 IU dosage regimen with a three-month interval, to 15 pediatric patients, without the “patient-specific follow-the-pain” protocol in their 24-week double-blind, placebo-controlled, randomized crossover study, which was succeeded by a subsequent 24-week open-label phase where all the participants were administered OnabotulinumtoxinA [28]. Winner et al.’s RCT represented the largest investigation, enrolling 125 children and adolescents. The participants were administered a single dose of either 155 IU or 74 IU of OnabotulinumtoxinA or placebo injections, with a subsequent 12-week follow-up period [29]. The remaining studies had a variety of administrating protocols. The retrospective analyses conducted by Karian et al. and Goenka et al. offered 155 IU of OnabotulinumtoxinA at three-month intervals, following the strict PREEMPT injection protocol; Karian et al. provided two treatment cycles, whereas Goenka et al. delivered three to four cycles of OnabotulinumtoxinA [32,35]. In the study of Kabbouche et al., 100 IU were initially administered in fixed injection sites (frontalis, procerus, corrugators, and temporalis areas), and if the procedure was tolerated, a second dose was administered at twice the initial quantity after a two-month interval [36]. In the case series analyzed by Ahmed et al. and Chan et al., a protocol involving lower doses, specifically of the order of 100 IU was implemented [39,40]. In one study, ultrasound-guided injection of active muscular trigger point was performed, across one to four distinct muscle sites, with a smaller total dosage per session, which ranged from 20 IU to 90 IU of OnabotulinumtoxinA [41]. All the study characteristics are shown in Table 1.

### 2.2. Participant Characteristics

Most of the pediatric participants across the included studies were adolescents, indicated by a mean age of 15.4 years and an age range spanning from 11 to 18 years [28,29,30,31,32,33,34,35,36,37,39,40,41]. Exceptionally, one study expanded its inclusion criteria to encompass children as young as 8 years old [38]. The studies demonstrate a marked predominance of female participants, with their representation exceeding 70% in most cases and frequently surpassing 80% [28,29,30,31,32,33,34,35,38,39]. The average age of migraine onset was documented in only three studies, spanning from 10.5 to 12.8 years of age [29,32,33]. Among the five studies that included race as part of their demographic data collection, a notable trend emerged indicating a predominant representation of non-Hispanic whites, constituting over 60% of the sampled pediatric population [28,29,30,32,38]. Depression, anxiety, and sleep disorders were the most common comorbidities associated with chronic migraine [28,30,33,38,39]. The majority of pediatric and adolescent participants deemed eligible for treatment with OnabotulinumtoxinA had previously undergone at least two oral preventive treatments [24,28,30,31,33,35,38,39]. Similarly, most research studies incorporated pediatric individuals diagnosed with chronic migraine, adhering to the diagnostic criteria for headache as defined by the ICHD-3 [2]. However, these studies often did not detail additional migraine-specific features, such as the occurrence of aura. A retrospective analysis involving 34 pediatric patients characterized the headache distribution, noting holocephalic pain in 58.8% and bitemporal pain in 17.6% of the patients. Furthermore, the perception was characterized as throbbing in 70.6% and pulsating in 14.7% of instances [35]. The case series conducted by Ahmed et al. encompassed not only children and adolescents diagnosed with chronic migraine but also featured two cases of new daily persistent headache, two cases of chronic tension-type headache, and one case of trochlear neuralgia [39].

### 2.3. Efficacy

The efficacy of OnabotulinumtoxinA in pediatric migraine was measured based on a spectrum of parameters, comprising both subjective and quantitative evaluations of migraine attack severity, frequency, duration, and the impact on the quality of life. Most of the studies showed a significant statistical decrease in the headache frequency, measured in mean monthly migraine days (MMDs), with reductions ranging from 6.1 to 17.5 days amongst the studies [28,31,32,33,34,35,36,37,38,41]. Winner et al.’s RCT stood out as the only investigation that failed to demonstrate a statistically significant decrease in MMDs between the OnabotulinumtoxinA groups, treated either with 155 U or 74 U, and the placebo group [29]. While Winner et al.’s study did not demonstrate a statistically significant variance between subgroups regarding a 50% reduction in the migraine episode frequency after a single treatment cycle, three retrospective analyses revealed that 55.8% to 73.5% achieved a 50% reduction in headache frequency after one to four treatment cycles of OnabotulinumtoxinA [29,31,33,35]. Eight studies assessed the severity of migraine attacks using the visual analog scale (VAS) [42], an intensity psychometric response scale employed to quantify an individual’s pain intensity, represented as a continuum from 0 (no pain) to 10 (worst imaginable pain) [28,32,34,35,36,37,38,41]. Only in one study was the observed reduction in pain intensity not found to be statistically significant; however, it should be noted that it is the only study where the mean baseline score was lower than in the others (VAS < 6) [36]. Two studies highlighted the correlation between additional treatment cycles and a reduction in both the frequency and severity of migraine attacks. The MMDs declined from 16.6 and 18.9 after the first cycle of OnabotulinumtoxinA to 9.5 and 7.6 after the fourth cycle, respectively. Concurrently, the percentage of patients experiencing severe migraines decreased from 41.8% to 16.2% during this interval, and the VAS score decreased gradually from 8.8 to 3.4 [33,35]. Concerning the duration of individual migraine attacks, a statistically significant reduction was exclusively demonstrated by one retrospective analysis following the administration of OnabotulinumtoxinA [38].

In a single study, acute medication reduction was assessed; with OnabotulinumtoxinA therapy, there was a reduction in the frequency of abortive medication usage among patients. Patients experienced a reduction from an average baseline of 14.6 medications per month to five medications after the completion of the fourth treatment cycle [35]. OnabotulinumtoxinA was associated with an improved quality of life, as measured either via the PedMIDAS score or via other questionnaires, such as the migraine-specific quality of life instrument and the Headache Impact Test 6 (HIT-6). [28,30,36,40,43]. However, once more, in the study conducted by Winner et al., no statistically significant distinction was observed regarding the PedMIDAS scores between the treatment and placebo groups [29]. Only two research groups evaluated the impact of OnabotulinumtoxinA treatment on the requirement for hospitalization. Both studies revealed a statistically significant reduction in the necessity for emergency department visits and admission [28,35].

### 2.4. Safety and Tolerability

OnabotulinumtoxinA administration appears to be generally well tolerated in the pediatric population, with reported treatment-related side effect rates ranging from 0% to 47% [28,29,30,31,32,33,34,35,36,37,38,39,40]. Schroeder et al.’s case series’ study, which was characterized by a notably lower dosage of OnabotulinumtoxinA and fewer injection sites, stands as the singular study devoid of reported adverse events [41]. In the remaining studies, the predominantly reported side effects were mild and primarily injection site related; they included neck pain, muscle weakness, flu-like symptoms, and a transient exacerbation of headaches. While Winner et al.’s RCT comprehensively detailed the observed side effects, which were documented in 20% of the patient cohort, Shah et al.’s RCT did not conduct any analysis of the undesirable effects [28,29]. No serious adverse events were documented in either the RCTs or the retrospective studies. Eyelid edema and inflammation, treated with oral antibiotics, were reported by Kabbouche et al. [36]. Treatment discontinuation was noted in two studies [33,35]. In the first study, a discontinuation rate of 19.5% was attributed to the non-tolerance to the injections, with a noteworthy statistically significant trend observed for treatment cessation among younger children. In the second study, a 14% discontinuation was attributed to adverse effects, although these were not subjected to further analysis [33,35].

## 3. Discussion

Migraine in the pediatric population frequently results in substantial functional impairment across home, school, and social domains, often leading to absences from school or reduced productivity. School functioning is crucial as children and adolescents with chronic pain conditions exhibit higher rates of school absenteeism, diminished academic achievement, heightened distractibility in class, and elevated susceptibility to depression compared to their healthy counterparts of a similar age [44,45]. In addition, studies indicate that children with chronic migraine often continue to present the same form of migraine in adulthood [46]. Therefore, prompt diagnosis and intervention are imperative for optimizing outcomes in children and adolescents. The American Academy of Neurology (AAN) and the American Headache Society (AHS) released in 2019 a guideline summary for pediatric migraine prevention [9]. Lifestyle and behavioral factors should always be addressed, as obesity, caffeine, alcohol, and tobacco use contribute to the appearance of frequent headaches [47,48]. Prophylactic interventions are recommended for children and adolescents experiencing recurrent headaches or disability attributable to migraines [9]. Among the pharmaceutical options commonly employed in such cases are topiramate, amitriptyline, valproic acid, and propranolol. It is important to highlight that the most extensive comparative efficacy investigation concerning preventive pharmacotherapy for pediatric migraine, namely the Childhood and Adolescent Migraine Prevention [11] trial, was terminated prematurely due to futility after interim findings indicating that amitriptyline and topiramate did not demonstrate superiority over a placebo in reducing headache occurrences [11].

Our systematic review provides compelling evidence suggesting that OnabotulinumtoxinA may represent a safer and more efficacious preventative treatment option for pediatric chronic migraine. Most of the studies included in the systematic review demonstrate that the use of OnabotulinumtoxinA reduces both the frequency and severity of migraine attacks. There appears to be an improvement in the quality of life in pediatric patients after the initiation of OnabotulinumtoxinA treatment. However, nearly all the studies exhibited a variance in the adherence to the PREEMPT protocol, complicating the consolidation of results for the formulation of reliable and conclusive interpretations. Furthermore, one of the two clinical trials failed to demonstrate the superiority of OnabotulinumtoxinA administration over a placebo in mitigating the frequency and severity of pediatric migraine [29]. Winner et al.’s RCT, despite encompassing a larger cohort of children and adolescents, employed a three-armed design, however with balanced demographic sub-groups [29]. Conversely, Shah et al.’s trial adopted a crossover design, wherein each patient served as their own control, and offered three treatment cycles over a quadruplicate follow-up period of 48 weeks, in contrast to Winner et al.’s single treatment cycle lasting 12 weeks [28,29]. While Winner et al. reported no statistically significant decrease in the frequency or intensity of migraine attacks, the absence of baseline scores is crucial for understanding the initial characteristics of the migraine attacks, underscoring the importance of this information [29]. Concerning outcomes, certain articles employed qualitative methodologies, while others utilized standardized assessment tools such as PedMIDAS and VAS, which are endorsed by both the American Neurological Association (ANN) and the American Headache Society (AHS) for evaluating pediatric migraine and guiding the treatment strategies [9]. OnabotulinumtoxinA demonstrates favorable tolerability within the pediatric population but is not entirely devoid of risks. Notably, in one retrospective analysis, the requirement for multiple injections led to treatment discontinuation in younger children and there is one case report of progressively declining pulmonary function attributed to OnabotulinumtoxinA, which gradually restored after treatment discontinuation [33,49].

OnabotilinumtoxinA has been employed for over three decades in managing spasticity in pediatric patients diagnosed with cerebral palsy [50]. Initial clinical applications typically involved a conservative dosing strategy, administering approximately 2 U/kg of body weight [51]. However, recent advancements in treatment protocols have embraced a more comprehensive approach, targeting multiple muscle groups [52,53,54]. This contemporary methodology has demonstrated the safety of administering substantially higher doses, even up to threefold the maximum dosage recommended for chronic migraine therapy within a single session [52,53,54]. These findings suggest a favorable safety profile for higher dosages of botulinum toxin in the context of pediatric spasticity management [52,53,55].

The conclusions drawn from this systematic review illuminate the considerable potential of OnabotulinumtoxinA as an effective therapeutic strategy for mitigating the impact of migraine in pediatric populations. However, to fully understand and evaluate its therapeutic benefits, there is an emerging need for future research endeavors to focus on several critical areas.

Firstly, there is a distinct lack of standardization regarding the outcomes measured across different studies, making it challenging to compare the results directly and to draw definitive conclusions about the efficacy of OnabotulinumtoxinA in a pediatric population. Future research should aim to establish a uniform set of eligibility criteria, and primary outcome measures, potentially including the reduction in monthly migraine days, improvements in quality-of-life scale scores, and decrease and responsiveness in acute medication use, to facilitate more meaningful comparisons and meta-analyses. To that extent, the International Headache Society has published guidelines for controlled trials of preventive treatment of migraine in children and adolescents, which should be taken into consideration when designing and conducting clinical trials [56,57].

Secondly, the methodologies surrounding injection protocols—such as the specific muscle sites targeted, the doses administered, and the intervals between the treatment sessions—vary significantly between the existing studies. This variability highlights the complexities in determining the most effective treatment regimen. Future studies should strive to detail the injection protocols used meticulously and investigate the impact of the different protocols on the treatment outcomes.

Moreover, understanding the long-term effects and safety profile of OnabotulinumtoxinA in pediatric patients remains paramount. While the initial findings are promising, comprehensive studies that monitor patients over extended periods are essential to ensure the enduring safety and effectiveness of this treatment modality.

## 4. Limitations

This systematic review presents various limitations. Among the 14 studies included, merely two were RCTs. The consistency of doses and treatment cycles varied significantly both across studies and within individual studies, with doses ranging from 20 IU to 200 IU and cycles ranging from one to four. Additionally, there was inconsistency in the parameters measured across the studies. While most studies reported on the frequency and severity of migraine episodes, there was a lack of specific mention regarding the utilization of acute therapeutic approaches or the impact of OnabotulinumtoxinA on the accompanying migraine symptoms.

## 5. Conclusions

OnabotulinumtoxinA holds significant promise as a treatment option for children and adolescents with migraine. To advance this field, future studies are needed to standardize the treatment outcomes, refine the injection protocols, and explore the long-term implications of this treatment, that carries the lowest risk for adverse events compared to standard anti-migraine pharmaceutical options currently available for the pediatric population.

## 6. Methods

This review was conducted in accordance with the Preferred Reporting Items for Systematic Reviews and Meta-Analysis (PRISMA) guidelines (accessed on 15 December 2023) [58]. No institutional review board approval was required since this review is based on previously published data, and it did not include individual patient data. The PRISMA flow diagram and checklist is presented in Figure 1 and Appendix A, respectively.

### 6.1. Eligibility Criteria

The inclusion criteria were studies that recruited individuals with chronic migraine and aged younger than 18 years old who were treated with OnabotulinumtoxinA, as no studies analyzing other toxins in pediatric migraine have been identified. Case series, cohorts (retrospective and prospective), case-control, cross-sectional, and randomized controlled trials (RCTs) were included in this review. Finally, we excluded case reports, reviews, descriptive, animal, and in vitro studies. No limits were applied to gender or race. Only English-language articles were considered.

### 6.2. Information Sources and Search Strategy

The literature search was conducted by systematically searching Medline (via PubMed) and Scopus on the 10th of February 2024 using the following search string: (botox OR Botulinum toxin OR Botulinum toxin A OR onabotulinumtoxina OR botox toxin OR onabotulinum toxin a) AND (migraine OR headache OR chronic migraine) AND (pediatric* OR paediatric* OR child* OR adolescen* OR teenage*). In addition, the full reference lists of the retrieved studies were also searched to identify additional articles (the “snowball” technique).

### 6.3. Selection Process

The selection of studies was conducted through a methodical, three-phase sequential approach. In the initial phase, the titles and abstracts of all electronic records were systematically reviewed to ascertain potentially eligible studies. In the subsequent phase, articles considered provisionally eligible were acquired in their entirety for further evaluation. In the final phase, studies that did not report the outcomes of interest or that met any exclusion criteria were systematically excluded from consideration. This process of study selection was independently carried out by two researchers (A.M. and A.R.), with any discrepancies being addressed by a third, independent researcher (T.M.).

## Figures and Tables

**Figure 1 toxins-16-00295-f001:**
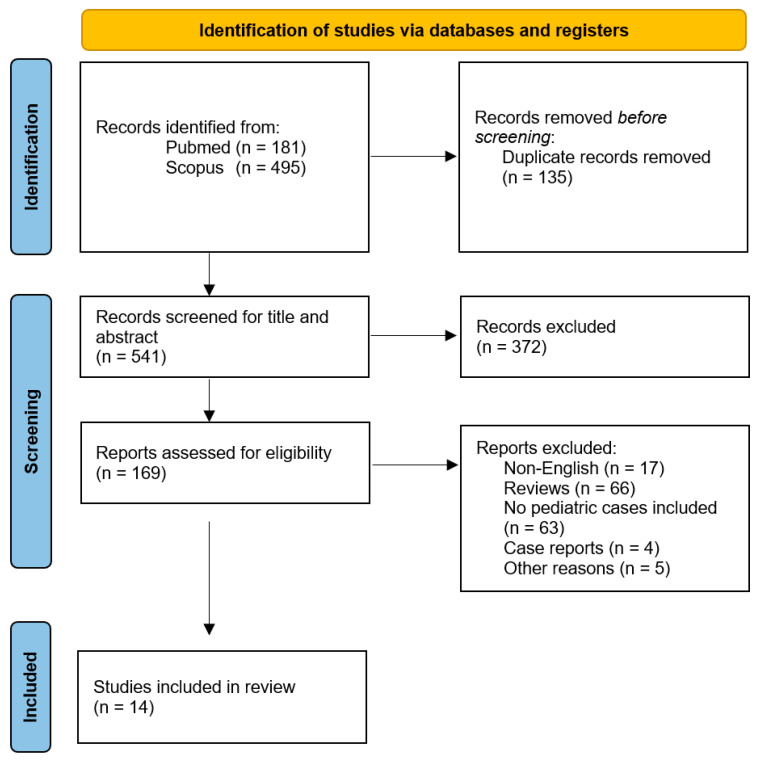
Flow diagram of study selection.

**Table 1 toxins-16-00295-t001:** Comprehensive overview of included studies in the systematic review.

Study	Number of Participants	Mean Age in Years (Range)	Dose	Protocol	Efficacy—Primary Endpoints	Treatment-Related AE
Shah et al. [28] RCT	15	15 * (14–16)	1st phase: 155 IU or placebo 2nd phase: 155 IU for 2 cycles	injections every 12 weeks, fixed 31-site injection PREEMPT protocol	median MMD^5^ 28 → 20 (*p* = 0.038) medianPNRS^6^ 8 → 5 (*p* = 0.047) median PedMIDAS^7^ 4 → 3 (*p* = 0.047) median duration 24 h → 10 h (*p* = 0.148)	No serious AE reported
Winner et al. [29] RCT	125	15.1	155 IU or 74 IU vs. placebo single cycle	single injection fixed 31-site injection PREEMPT protocol	no significant difference in MMD, mean duration per episode, PedMIDAS	total: 20% (neck pain, nasopharyngitis, musculoskeletal pain, migraine, dizziness) serious: 0
Akbar et al. [30] retrospective analysis	24	15.4 * (12–17.5)	155–200 IU for 2 to 4 cycles	fixed 31-site injection PREEMPT protocol + patient-specific follow-the-pain protocol	median HIT score 71.5 → 53 (*p* < 0.001) median PedMIDAS 4 (67.5) → 2(17.5) (*p* < 0.001)	total: 20% (neurologic, gastrointestinal, laboratory changes, renal)
Horvat et al. [31] retrospective analysis	32	16 (13–17)	155–200 IU (patient-specific follow-the-pain protocol)	modified 31-site injection PREEMPT protocol	mean MMD 24 → 9.1 (*p* < 0.001) 68% had a 50% reduction in headache frequency	total: 4% (neck pain, headache)
Karian et al. [32] retrospective analysis	32	16 (13–17)	155 IU for 2 cycles	fixed 31-site injection PREEMPT protocol	mean MMD 21 → 13.6 (*p* < 0.001) mean PNRS 7 → 5.5 (*p* = 0.003) mean duration 18.5 h → 16 h (*p* = 0.14)	total: 47% (transient worsening of pain, flu-like symptoms, fatigue)
Papetti et al. [33] retrospective analysis	43	14.7 (12–17)	155–195 IU (patient-specific follow-the-pain protocol) for 4 cycles	fixed 31-site injection PREEMPT protocol + patient-specific follow-the-pain protocol	mean MMD 21.8 → 9.5 (*p* < 0.05) severe attacks 41.8% → 16.2% 55.8% had a 50% reduction in headache frequency	total: 32% (injection-site edema, pruritus, headache, neck pain, muscle weakness) serious: 0 discontinuation 19.5%: non-tolerance to the injections
Ali et al. [34] retrospective analysis	30	16.5	155–185 IU (patient-specific follow-the-pain protocol)	modified 31-site injection PREEMPT protocol + patient-specific follow-the-pain protocol	mean MMD 24.4 → 14.8 (*p* < 0.001) mean VAS 7.5 → 4.3 (*p* < 0.001)	total: 3.3% (nausea) serious: 0
Goenka et al. [35] retrospective analysis	34	17.5 (12–21)	155 IU for 4 cycles	fixed 31-site injection PREEMPT protocol	mean MMD 18.9 → 7.6 (*p* < 0.001) mean PNRS 8.3 → 3.4 (*p* < 0.001) mean use of abortive medications 4.3 → 1.4 (*p* < 0.001) 73.5% had a 50% reduction in headache frequency	discontinuation 14%: AE (lateral eyebrow elevation, severe pain during injection)
Kabbouche et al. [36] retrospective analysis	45	16.8 (11–21)	1st dose 100 IU 2nd 200 IU after 2 mo	fixed 31-site injection PREEMPT protocol + patient-specific follow-the-pain protocol	mean MMD 27.4 → 21.3 (*p* = 0.009) mean PedMIDAS 55.7 → 32.4 (*p* = 0.19) mean mean VAS 5.4 → 4.6 (*p* = 0.14)	total: 6.1% (injection-site pain, myalgia, eyelid inflammation, eyelid pain, eyelid edema)
Santana et al. [37] retrospective analysis	65	15 (11–18)	155–200 IU (patient-specific follow-the-pain protocol) for 1 cycle	fixed 31-site injection PREEMPT protocol + patient-specific follow-the-pain protocol	mean MMD 16 → 4 mean VAS 7.5 → 2.3	total: 3.1% (dizziness, neck pain, flu-like symptoms)
Shah et al. [38] retrospective analysis	10	15 (8–17)	155–215 IU (patient-specific follow-the-pain protocol) for 1 to 11 cycles	modified 31-site injection PREEMPT protocol + patient-specific follow-the-pain protocol	mean MMD 15 → 4 (*p* < 0.001) mean VAS 6 → 4 (*p* = 0.006) mean duration 8 h → 0.8 h (*p* = 0.025)	total: 22.8% (injection-site pain, lower extremity weakness)
Ahmed et al. [39] case series	10	15.5 (11–17)	100 IU for 1 to 3 cycles	fixed 31-site injection PREEMPT protocol	no statistical tests 40% had a reduction in frequency and intensity	total: 30% (flu-like symptoms, paraesthesia) serious: 0
Chan et al. [40] case series	6	(14–18)	100 IU for 1 to 9 cycles	modified 31-site injection PREEMPT protocol	no statistical tests 25% improvement in the total scores of the MSQOLi	injection-site hematoma, injection-site numbness, mild ptosis
Schroeder et al. [41] case series	5	13 (10–16)	20–90 IU	Ultrasound-guided injection of active muscular trigger point	no statistical tests mean MMD 20 → 5 mean VAS 6.6 → 2.4	total: 0

AE: adverse events, RCT: randomized control trial, IU: international units, VAS: visual analog scale, HIT: headache impact test, PREEMPT: phase 3 research evaluating migraine prophylaxis therapy, MMD: monthly migraine days, PNRS: pain numeric rating scale (0 to 10), PedMIDAS: pediatric migraine disability assessment score, MSQOLi: migraine-specific quality-of-life instrument; * Median.

## Data Availability

The data supporting this systematic review were derived from previously published studies, which have been duly cited.

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
