# Peer review of "Onabotulinumtoxina in the Prevention of Migraine in Pediatric Population: A Systematic Review"

_toxins, 2024, doi:10.3390/toxins16070295_

Round 1

Reviewer 1 Report

Comments and Suggestions for Authors

This is a systematic review addressing an important topic. There is insufficient data on migraine prophylactic treatments in the paediatric age group, so this review is welcome.

It is encouraging that there are some indications of efficacy for botulinum toxin in this group.

Of course the main concern in treating children is safety. Adverse effects from botulinum toxin may be local (and would be specific for the sites of injection) or systemic (in which case the total dose would be the most important factor). The paper would be enhanced by referencing some of the data that is already available regarding systemic effects of toxin therapy for cerebral palsy in paediatric patients (and the dosing recommendations for CP derived from this data). This would help put the doses used in treating migraine into context and would support the view that systemic toxicity is unlikely with typical protocols used for migraine.

Author Response

Thank you very much for your review. As per your suggestion, available data on the usage of onabotulinumtoxinA in cerebral palsy have been included in the discussion section. We provided relevant data regarding OnabotulinumtoxinA usage and dosage in cerebral palsy in pediatric patients (Lines 242-251), to let the readers have a more insightful view regarding the comparison of the dosages used in those two pathologies and to highlight the safety of the use of OnabotulinumtoxinA.

Reviewer 2 Report

Comments and Suggestions for Authors

    Botulinum type A has been approved as a preventive treatment for chonic migraine in adult patients who have not responded adequately or are intolerant to migraine preventive drugs. The clinical practice guidelines on headaches propose, following the publication of the PREEMPT studies, to initiate this treatment in chronic migraine patients with intolerance, contraindication or lack of response to at least two preventive drugs (a beta-blocker and topiramate) that will have been used at the minimum recommended doses and for at least 3 months. While botulinum toxin is approved and proven effective in adults with chronic migraine, some reports revealed that it is not effective in children and adolescents and the need for multiple injections often leads to treatment discontinuation in young children.

    This systematic review entitled "Onabotulinum toxin A in the prevention of migraine in pediatric population: a systematic review" provides evidence suggesting that botulinum type A may represent a safer and more efficacious preventive treatment option for pediatric migraine, but as the authors point out in their discussion that nearly all studies exhibited variance in adherence to the PREEMPT protocol, complicating the consolidation of results for the formulation of reliable and conclusive interpretations. Although the manuscript is well structured and the authors have documented the paper very well, the therapeutic option is already known and the manuscript does not represent any new insight into the entity.

Author Response

Thank you very much for your constructive review of our manuscript titled "Onabotulinum toxin A in the prevention of migraine in pediatric population: a systematic review." We appreciate your recognition of the structured presentation and comprehensive documentation of our work.

We would like to emphasize the necessity of our article in addressing a critical gap in the current literature on chronic migraine treatment with OnabotulinumtoxinA, particularly in pediatric populations. While it is acknowledged that botulinum toxin type A is a known therapeutic option for adult population, our systematic review highlights an essential aspect that has not been sufficiently addressed: the heterogeneity in clinical trials and adherence to the PREEMPT protocol.

The PREEMPT protocol is a standardized guideline that has demonstrated efficacy and safety in adult populations. However, our review identifies significant variance in its application across pediatric studies. This heterogeneity in trial design and protocol adherence complicates the consolidation of results and impedes the formulation of reliable, conclusive interpretations. Scientists and clinicians must be aware of these variations to critically evaluate existing evidence and to advocate for better-designed, more homogeneous studies in future research. Addressing this unmet need is crucial for advancing our understanding and improving the reliability of clinical recommendations.

Furthermore, our review underscores the safety profile of OnabotulinumtoxinA in pediatric populations, even in the context of these diverse study designs. Despite the protocol variances, the safety data remains consistently favorable, reinforcing the potential of onabotulinumtoxinA as a viable preventive treatment for pediatric migraine. This aspect of our findings is particularly significant for clinicians considering therapeutic options for younger patients, where safety concerns are paramount.

Although we do not present any new data as this is a systematic review and not an original study, this SR compiles and analyses all the current literature on the subject. We aim to inform and guide future research towards more rigorous and standardized study designs, ultimately leading to more definitive and actionable clinical insights. We believe this contribution is valuable and necessary for the ongoing development of effective migraine treatments in pediatric populations.

Thank you once again for your constructive feedback. We hope this response clarifies the significance and necessity of our work.

Reviewer 3 Report

Comments and Suggestions for Authors

1. It is not always clear to reader if this study in fact only includes chronic migraine patients. Why did the authors include any studies where the definition of chronic migraine was not specified? In this case it seems difficult to define them as chronic. Why does Ë‹chronic´ not appear in the heading? Do all studies in the table include only chronic migraine patients? Please highlight where this is not the case or unclear.

2. Many publications included are in fact case reports which lower the quality of the review. What is the rationale behind? Why did the authors not exclude them  a priori defining that approach in their selection criteria? This would strengthen the conclusion. 

3. Why did the authors only focus on onabotulinum toxin? Are there any other studies available using other types of the toxin for chronic migraine with series > 1. At least, this should be mentioned.  

4. 277. Exchange Botox by onabotulinum toxin.

Comments on the Quality of English Language

This review needs major revision. 

Author Response

  1. We thank the reviewer for the insightful comment. All studies had included patients with chronic migraine and this has been clarified, as per your suggestion (Lines 81-85).
  2. We appreciate the reviewer for the perceptive remark. As the amount of studies regarding the use of OnabotulinumtoxinA was small, we tried to gather as many patients as we could find, even though not many data were available to formulate and IPD Meta-analysis. As correctly mentioned this lowers the results and conclusion of the SR. To enhance the quality of the systematic review, case reports were excluded as per your suggestion.
  3. Thank you for the remarkable comments. OnabotulinumtoxinA is the only type of botulinum toxin tried and approved for the treatment of chronic migraine in adults. There are not any other studies that used other types of toxins for the treatment of chronic migraine and this is mentioned in the manuscript as pre your instructions.  (lines 64-68)
  4. Thank you for the comment. It has been corrected.

Round 2

Reviewer 2 Report

Comments and Suggestions for Authors

I would like to express my gratitude to the authors for taking the time to explain their thoughts. I do not really know how to revise the paper; I agree that this is an excellent review, but I feel that the article itself does not contribute anything particularly new or innovative to the field of childhood migraine treatment. As a reviewer, I have been kind enough to explain in detail why I felt this article was not entirely correct and why this type of research is not really useful. 

Reviewer 3 Report

Comments and Suggestions for Authors

.

Comments on the Quality of English Language

.